# Parental Satisfaction with Caregiving across the Life Span to Their Children with Developmental Disabilities: A Cross-Sectional Study in Iran

**DOI:** 10.3390/ijerph17051576

**Published:** 2020-02-29

**Authors:** Sayyed Ali Samadi, Roy McConkey, Ghasem Abdollahi Boghrabadi

**Affiliations:** 1Institute of Nursing and Health Research, Ulster University, Belfast BT37 0QB, UK; r.mcconkey@ulster.ac.uk; 2Department of Psychology, Payame Noor University, Tehran 19569, Iran; psy-abdollahi@pnu.ac.ir

**Keywords:** parental satisfaction, intellectual and developmental disabilities, caregiving, Iran

## Abstract

The increased life expectancy of adult individuals with developmental disabilities and the likelihood of parents having to continue caregiving into their old age is an emerging international issue which deserves investigation, especially concerning satisfaction with caregiving. Moreover, this needs to be assessed in different cultures in order to create a better understanding of how families are best supported in their lifelong caregiving. A sample of 408 parents was gathered in six cities across Iran with a son or daughter who had a confirmed developmental disability. Self-completed measures of satisfaction and stress were obtained along with demographic details of the child and family. Satisfaction with caring was generally positive and was similar for mothers and fathers, for older as well as for younger parents; and between different types of developmental disabilities. However, both personal and child satisfaction decreased when parents reported increased stress and when caring for teenage and adult offspring and those with behavior problems. Parents need to receive support to sustain their motivation and satisfaction with caregiving if their quality of life and that of their children with disabilities is to be maintained and enhanced across their lifespan.

## 1. Introduction

Internationally, most children with developmental disabilities (DDs) live with their parents in the family home. Furthermore, families continue to be the primary care providers for people with DDs well into adulthood, especially in less-affluent countries [1]. An increasing number of children with more complex disabilities now survive into adulthood and often have close to a regular lifespan [2]. Although there are some reports of improvements in behavior and ability as people with developmental disabilities move from childhood into adulthood, most require some degree of supervision throughout their lives [3,4,5].

Around the world, increasing numbers of adults with developmental disabilities depend on ageing parents. For example, in the United States, more than 75% of adults with developmental disabilities live at home with their parents, and more than 25% of these caregivers are aged 60 years or older [6]. The World Health Organization [7] identified adults with developmental disabilities and their older caregivers as two vulnerable groups who have to face the challenges of the ageing process along with maintaining optimism in their caregiving role. Yet to date, the main focus of research into caregiving has been focused on the parents of children with developmental disabilities. As Bickenbach et al. [8] argued, the impact of ageing on caregivers needs to be assessed in order to better support families in their lifelong caregiving.

The available literature indicates that parents caring for adults with developmental disabilities have higher rates of poor physical health and mental health symptoms compared with their peers [9]. In a comprehensive review, it was concluded that these parents also report more stress and less satisfaction with caregiving [10]. However, few studies have investigated caregiving in less-affluent, non-Western countries and even fewer studies have taken a lifespan perspective involving older parents of adult persons with developmental disabilities.

As a preliminary study in Iran, a scale for assessing parental satisfaction with caregiving was developed and validated [11]. Parents of children with developmental disabilities, particularly those whose children had Autism Spectrum Disorders (ASD), demonstrated lower levels of parental satisfaction with caregiving compared with parents of typically developing children. However, to our knowledge, few studies outside Western countries have examined parental satisfaction with caring for adult sons and daughters.

Most of the focus in past research has been on mothers. However, the contribution of fathers may be especially pertinent in certain cultures, particularly in patriarchal societies. Previous research suggests that fathers of children are similarly affected [12] although not to the same extent as mothers which might be due to the fact that fathers of this group of children are rarely included in research on the impact of caregiving [13]. However, an Iranian study found that mothers had poorer health and wellbeing than fathers, although they did not differ in terms of stress and satisfaction with caregiving [14]. It is not known if these findings hold with parents caring for teenagers or adults with developmental disabilities.

The present study addressed the following research questions based on the foregoing literature review and the identified gaps in our knowledge: Are there differences between young, middle-aged and older parents with respect to their satisfaction with caregiving? Is there a link between parental satisfaction with caregiving and the types of stress they experience? Do mothers and fathers differ in their sources of stress and satisfaction with caregiving? Does the nature of their child’s developmental disability affect their satisfaction with caregiving?

### Services in Iran

In Iran, preschool children with disabilities and those deemed to be ineligible for schooling because of the severity of their disability attend day care centers under the aegis of the Iranian State Welfare Organization (ISWO). Vocational training centers are also provided by the ISWO for young adults. These include privately run centers as well as those directly managed by the ISWO. Fees are paid by the ISWO for more disadvantaged families.

The ISWO currently has over 500 approved centers across the country with over 20,000 children and young adults enrolled in them with various intellectual disabilities including ASD and Attention Deficit and Hyperactivity Disorder (ADHD). All center staff hold a university degree (two-year course) and some will have a postgraduate, two-year diploma. The centers operate five mornings a week from 8:00 am to noon with extra occupational, speech and language therapy, and educational sessions available in the afternoons at parental expense.

## 2. Materials and Methods

### 2.1. Participants

In all, 408 parents were recruited who had a child with a confirmed developmental disability and who had registered to the Iranian State Welfare Organization (ISWO). All were caring for their child with developmental disabilities at their home at the time of the study and one parent per household was recruited. The recruitment was done from different ISWO services for children and young adults with developmental disabilities in six cities across Iran.

Table 1 summarizes the demographic characteristics of the parents and children from information provided by the parents. Information was also sought on any additional problems the parents experienced with the child along with problems in behavior management.

### 2.2. Measures

The Parental Satisfaction with Caring for a child with Developmental Disability Index (PSCDDI) [11]: This scale was developed and validated in Iran specifically with parents of persons with developmental disabilities. It consists of six items indicative of personal satisfaction of parents and six items related to child satisfaction. Each item is rated on a five-point Likert scale and a summed score can be calculated by adding responses across the six items for personal and for child satisfaction. Higher scores are indicative of less satisfaction. The Cronbach’s alphas for these subscales were 0.883 for the personal items and 0.857 for the child items. The Pearson product-moment correlation between the two scores was r = 0.677 which, although significant (*p* < 0.01), indicates less than 50% of shared variance between the two subscales.

Parenting Stress Index (PSI) [15]: This widely used 36 item scale consists of items related to the stress experienced by parents in caring for the child and those relating more to parental distress. The first 33 items are rated on a five-point Likert scale from “Strongly Disagree” to “Strongly Agree”, with three items also rated into one of five options. In the present study with 408 Iranian parents, a factor analysis of the 36 items broadly replicated the three subscales in the original standardization study in the USA, which other studies have confirmed (e.g., [16]). Total scores were then calculated for three subscale items: (1) Parental Distress (12 items); (2) Difficult Child (13 items); (3) Parent–Child Dysfunctional Interactions (11 items). A total score can also be obtained by adding the three subscale scores. Higher scores on all measures are indicative of greater stress.

The Cronbach’s alpha for the “Parental Distress” items was 0.842, for the “Difficult Child” items it was 0.853, and for “Parent–Child Dysfunctional Interactions” it was 0.788. The Pearson product-moment correlations between the three scores ranged from r = 0.507 to r = 0.719 which although significant (*p* < 0.001) indicates less than 50% of shared variance between the three subscales. Unsurprisingly, each subscale correlated highly with the total scores, with correlations ranging from 0.816 (Difficult Child) to 0.872 (Parental Distress).

A questionnaire requesting demographic information about the child and family was also prepared (see Table 1).

All the questionnaires were made available in Farsi and had been used in previous studies in Iran.

### 2.3. Ethical Considerations

A formal ethical opinion was not sought for this study, but rather approval was sought from the ISWO as is common practice in Iran. Parents were assured that all information was confidential to the researchers and they would not be identified in any reports. Moreover, they could choose not to answer any questions. They indicated their consent by completing the questionnaires and returning them to the centers. Declining to participate in the study did not affect the services that they or their relative received from the ISWO.

### 2.4. Data Collection Procedure

Following approval for the study from the ISWO, 24 daycare centers and vocational training centers in six cities across Iran were approached. The administrator of each center was first sent an official letter from the ISWO central bureau about the study. The officials in each center were further asked to inform parents about the purpose of the study in the monthly meeting sessions they held for parents. The centers were sent copies of the questionnaires with an information letter about the study containing assurances of confidentiality as noted above. Parents willing to participate completed the questionnaires either at the meeting or at home. In the case of the latter, they returned them to the center in a special folder from where they were collected by the project staff.

In all, 408 completed questionnaires were returned. Although no accurate record was made of the number of parents who had been approached to take part in the study, an estimated response rate of 70% can be calculated based on the number of persons enrolled in the 24 centers—around 600 in all.

## 3. Results

Table 2 presents the mean scores across all parents on the five measures that were the focus of the study, along with details of the minimum and maximum scores. The satisfaction measures tended to be skewed towards greater satisfaction whereas the stress scores were distributed normally.

Table 3 summarizes the correlations among the five measures of satisfaction and stress in order to identify the relationships between the subscales. Although all correlations were statistically significant, they were higher on the subscales within the stress and satisfaction measures than they were between the two measures of satisfaction and the three measures of stress. Those parents who reported more stress were also less satisfied with caregiving.

Bivariate analyses using one-way analysis of variance tests were undertaken between the demographic variables listed in Table 1 and the five parental measures of stress and satisfaction listed in Table 2. The significance level was set at *p* < 0.01 given the large number of comparisons that were made.

There were no significant differences in any of the five measures in terms of parent gender, whether they owned or rented their home (a proxy for socio-economic status), the birth order of the child with developmental disabilities, and whether the child had other problems.

However, on one or both of the satisfaction scores, parents of girls and those with children aged 13 years and over were significantly less satisfied with caregiving. Parents of children with ASD were less satisfied than those of children with learning disabilities, who in turn were less satisfied than those of children with ADHD. Furthermore, parents living in Tehran were less satisfied than those living in other cities. Similarly, parents aged 55 years and over had significantly less child satisfaction, while parents of children with behavior problems had less personal satisfaction.

Four demographic variables were significantly related to one or more of the parenting stress measures. Parents of girls and those with children aged 13 years and over had higher stress scores. Likewise, parents of children with ASD were more stressed than those of children with learning disabilities, who in turn were more stressed than those of children with ADHD. Furthermore, parents living in Tehran reported more stress than those living in other cities.

In order to account for the interrelationships among these variables, those that were significantly related to the satisfaction and stress measures were entered into a linear regression analysis. However, in the regressions for the three measures of stress, only the satisfaction scores were significantly predictive of stress. None of the other variables identified in the bivariate analyses described above added significantly to the regression.

By contrast, on the measures of satisfaction, other variables in addition to stress scores did contribute to the regression. For simplicity, this analysis is reported for the total satisfaction scores as this replicated the regression equation for the two subscales. As Table 4 shows, the parents who were less satisfied with caregiving were those whose children were aged 13 years and over compared to those 12 years and under, who reported the child having behavior problems, and who had reported more stress on the “Difficult Child” subscale of the PSI, but not on the other two subscales. Parents living in Tehran also reported less satisfaction, as did parents of girls, although the latter just failed to add significantly to the regression. These predictors accounted for 30% of the variance in satisfaction ratings.

## 4. Summary of Main Findings

Satisfaction with caring was generally positive and was similar for both mothers and fathers, for older as well as for younger parents, and between different types of developmental disabilities. However, both personal and child satisfaction decreased when caring for teenage and adult offspring (13 years and over) and when caring for those with behavior problems. Parents who reported higher stress arising from difficulties with their child reported less satisfaction. Parents living in Tehran and those with daughters also tended to report less satisfaction.

## 5. Discussion

The present study had a number of novel features. Fathers as well as mothers were recruited, and the scarcity of information about fathers is globally recognized [17]. Our findings suggest that Iranian fathers report similar patterns of satisfaction and stress as mothers do, which is contrary to some studies in Western countries [18] but confirms the findings from previous studies undertaken in Iran [14]. In part, this may arise from cultural differences where Muslim males are expected to assume a paternalistic and protective role in the family, whereas in most Western countries this is seen as a maternal role. Nevertheless, for mothers as well as fathers, child-related stress in particular seemed to have a negative impact on satisfaction with caregiving, as studies in other countries have also confirmed. For example, Hassall, Rose, & McDonald [19] reported that regardless of the child’s age or type of developmental disability, parenting stress is negatively associated with caregiving satisfaction.

The study also incorporated children and young adults with different confirmed disabilities who attended the same centers provided by the ISWO. Previous literature has suggested that parental satisfaction and stress vary according to the diagnostic label given to the child (e.g., [20]). However, once differences arising from child behaviors were taken into account in the present study, the effect of the diagnostic label was no longer significantly related to satisfaction. Thus, interventions aimed at reducing parental stress and increasing parental satisfaction with caregiving could be shared across families caring for children with differing disabilities, especially if the focus is on managing the child’s behavior which parents find challenging [21].

This study was designed to explore the impact of lifelong caregiving which families have to provide to sons and daughters with disabilities. Two aspects could be examined and compared: older parents and older children with younger parents and younger children. With respect to the latter, parents of teenagers or young adults were less satisfied with caregiving than those with children. This may reflect the extra challenges parents face as their children seek greater independence and mature sexually.

By contrast, older parents reported similar levels of satisfaction to younger parents once the child’s age as well as other factors were taken into account, as the regression analyses suggested. This is encouraging given the extended period of caregiving that parents are required to give to sons and daughters with developmental disabilities, especially in less-affluent countries where alternatives to family caregiving are scarce or non-existent. Although this finding may represent an acceptance by parents of their responsibilities, this cannot be taken for granted. As parents age, they may encounter other difficulties that did not arise within this sample of parents or were not included as variables within this study, such as declining physical and mental wellbeing of the parents, the loss of a partner through death, and increased poverty through loss of income [22]. Further research is needed to explore the impact of such variables with parents older than those recruited to the present study. Moreover, the families in this study were arguably advantaged through their link with services, whereas families who receive no services may be less satisfied and more stressed than those in the present sample. However, such families are not easily recruited to participate in research projects.

As Iran is a multicultural country with different ethnic minorities, sample recruitment was extended to six cities in Iran. The results showed a significant location effect on parental ratings of satisfaction with caregiving, with those living in the capital city reporting less satisfaction. This is all the more surprising as parents in Tehran had a wider choice of services and fewer transportation issues compared to those living in smaller cities. This finding is a useful reminder that samples recruited from one city may not be representative of parents nationally. Future research should make greater attempts to recruit samples from different locations rather than drawing convenience samples from the capital city, which often seems to happen.

Although the reasons for the differences within a country are not easily identified, cultural variations may contribute. In traditional Middle Eastern cultures, there is an expectation on families to care for those who are vulnerable, and these beliefs may be stronger away from capital cities. Similarly, the tendency for parents to report less satisfaction from caring for girls rather than boys may reflect beliefs that girls require more care and protection, and even more so if a girl has a developmental disability. Sabih and Sajid [23] reported similar findings in their study with Pakistani parents which found that parents of females with ASD had higher stress scores compared to parents of male children with ASD. However, the present study is not an adequate test of the impact of child gender on parental satisfaction and stress given the relatively small number of girls recruited into it. However, the gender imbalance is also reflective of the population attending centers, which may be another manifestation of families being more protective of their daughters. Nonetheless, larger samples of families with daughters would help to overcome the possible lack of statistical power in the present study to examine the differences between caring for male and female offspring.

A further limitation of this cross-sectional study was that the relationships identified are not necessarily causal. Rather, longitudinal studies with the same sample of parents are needed to investigate whether reductions in stress due to children’s behavior, for example, leads to a concomitant increase in parental satisfaction with caregiving. Or conversely, if the decrease in satisfaction could be avoided with caregiving, which the present study suggests may arise as the child transitions to adulthood, if parents are actively supported to cope with challenges that arise with teenagers. However, longitudinal studies are rare, especially those based on large samples of participants that are needed to control the other variables related to parental satisfaction with caregiving that this study has identified. Nonetheless, it proved possible to undertake this study with minimal resources and the active co-operation of the centers and parents. Indeed, service personnel are well placed to undertake longitudinal monitoring of parents’ experience of their caregiving. However, this requires a refocusing of services away from attending to the child, towards taking responsibility for supporting families in their caregiving at home.

## 6. Conclusions

Given the increased life expectancy of adult persons with developmental disabilities and the likelihood of ageing parents having to continue their caregiving well into their old age, it is imperative that they receive the support needed to sustain their motivation and satisfaction with caregiving if their quality of life and that of their children with disabilities is to be enhanced. Hopefully this study will be a precursor and stimulant to further research into the needs of family carers in all countries, but especially in less-affluent countries where, to date, studies are lacking.

## Figures and Tables

**Table 1 ijerph-17-01576-t001:** The demographic characteristics of the sample (N = 408).

Parental Characteristics		Number	%
Gender of Parents	Female	256	62.7
	Male	152	37.3
Age of Parents	30–39 years	168	41.2
	40–54 years	111	27.2
	55+ years	129	31.6
Home	Owned	160	39.2
	Rented	248	60.8
City	Tehran	195	47.8
	Other City	213	52.2
Developmental Disability of Child	Learning Disability	185	45.3
	ASD	115	28.2
	ADHD	108	26.5
Gender of Child	Male	334	81.9
	Female	74	18.1
Birth Order	First Born	189	46.3
	Second and Later Born	219	53.7
Age of Child	4–12 years	128	31.4
	13–19 years	128	31.4
	20–43 years	152	37.3
The Child has Behavior Problems	Yes	176	43.1
	No	232	56.9
The Child has Other Problems	Yes	76	18.6
	No	331	81.1

ASD—Autism Spectrum Disorders; ADHD—Attention Deficit and Hyperactivity Disorder.

**Table 2 ijerph-17-01576-t002:** The mean, standard deviation, and range of scores on the five measures of stress and satisfaction.

Statistical Data	Parental Distress	Difficult Child	Parent–Child Interactions	Personal Satisfaction	Child-Related Satisfaction
Mean (SD) Range	36.70 (8.02) 17–55	42.39 (8.35) 21–60	34.11 (6.68) 19–52	11.79 (4.78) 6–26	10.31 (3.94) 6–29
Range of Possible Scores	12–60	13–65	11–55	6–30	6–30

**Table 3 ijerph-17-01576-t003:** Pearson product-moment correlations among the three subscales of the Parenting Stress Index (PSI) and the two subscales of Parent Satisfaction (N = 408).

	PSI Parent–Child Interaction	PSI Difficult Child	Personal Satisfaction	Child Satisfaction
PSI Parental Stress	0.719 **	0.507 **	0.291 **	0.245 **
PSI Parent–Child Interaction		0.531 **	0.326 **	0.275 **
PSI Difficult Child			0.509 **	0.490 **
Personal Satisfaction				0.677 **

** *p* < 0.01.

**Table 4 ijerph-17-01576-t004:** The variables predictive of total satisfaction with caregiving as determined by the linear regression (N = 408).

Model of the Predictive Variables	Unstandardized Coefficients		Standardized Coefficients			95.0% Confidence Interval for B	
	B	Std. Error	Beta	t	Sig.	Lower Bound	Upper Bound
**Stress—Difficult Child**	0.191	0.024	0.404	7.839	0.000	0.143	0.238
**Stress—Parent**	−0.007	0.030	−0.015	−0.247	0.805	−0.066	0.051
**Stress—Parent–Child Interaction**	0.018	0.036	0.030	0.493	0.622	−0.054	0.089
**Age of Child**	1.806	0.454	0.380	3.977	0.000	0.913	2.699
**Child Has Behavior Problems**	−1.104	0.366	−0.139	−3.020	0.003	−1.823	−0.385
**City**	0.888	0.382	0.113	2.324	0.021	0.137	1.639
**Gender of Child**	0.784	0.434	0.077	1.806	0.072	−0.069	1.637
**Developmental Disability**	−0.278	0.265	−0.058	−1.049	0.295	−0.798	0.243
**Parental Age Groups**	−0.270	0.334	−0.063	−0.808	0.420	−0.926	0.387
**(Constant)**	−1.358	1.671		−0.813	0.417	−4.643	1.926

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
