# Peer review of "Parental Satisfaction with Caregiving across the Life Span to Their Children with Developmental Disabilities: A Cross-Sectional Study in Iran"

_ijerph, 2020, doi:10.3390/ijerph17051576_

Round 1

Reviewer 1 Report

This paper investigates the satisfaction of parents who care for children with developmental disabilities. While this topic is well studied in more developed countries, it remains blank in less affluent countries like Iran. This paper fills this gap well. The authors have a very good data set, and perform a comprehensive data analysis. However, I am a bit concerned about the way they present their results. The survey results contain many items. The authors often begin their data analysis without explaining the definitions clearly. For example, in Table 2, the authors directly list the summary of 5 measurements without any explanation. What is difficult child? Readers can tell these measurements can be classified into stress scores and satisfaction scores. What is the difference between them? In Table 3, the authors present PSI Child-Interaction, etc. What is PSI? If it refers to the Parenting Stress Index, then it would be better to state this ahead. What is the purpose to analyze the correlation between different measurements in Table 3? While reading, I realize the authors have done a lot of work, but I cannot understand why they did this. I would suggest restructuring the paragraphs and make it friendly to read. Besides, the tables seem sloppy. For example, in Table 2, some entries have parentheses, while some do not have. In summary, I think this is an important study, but it would be better for the authors to polish the writing.

Author Response

  1. This paper investigates the satisfaction of parents who care for children with developmental disabilities. While this topic is well studied in more developed countries, it remains blank in less affluent countries like Iran. This paper fills this gap well. The authors have a very good data set, and perform a comprehensive data analysis. 

    R: Thank you for this affirmation.

  2. However, I am a bit concerned about the way they present their results. The survey results contain many items.  The authors often begin their data analysis without explaining the definitions clearly. For example, in Table 2, the authors directly list the summary of 5 measurements without any explanation. What is difficult child?

    R: The information contained in Table 2 is described in the text – section 2.2
  3. Readers can tell these measurements can be classified into stress scores and satisfaction scores. What is the difference between them? 

    R: These concepts are referred to in the introduction as well as described in section 2.2.

  4. In Table 3, the authors present PSI Child-Interaction, etc. What is PSI? If it refers to the Parenting Stress Index, then it would be better to state this ahead     R: The abbreviation PSI is given in section 2.2.
  5. What is the purpose to analyze the correlation between different measurements in Table 3?  

    R: The correlations identify the presence of relationships which is one of the research questions posed by the study.  Correlations are also the basis for the regression analyses which we undertook.

  6. While reading, I realize the authors have done a lot of work, but I cannot understand why they did this. I would suggest restructuring the paragraphs and make it friendly to read.   

    R: The research questions have been clarified and the paper structured accordingly in terms of methods, results and discussion. 

  7. Besides, the tables seem sloppy. For example, in Table 2, some entries have parentheses, while some do not have. In summary, I think this is an important study, but it would be better for the authors to polish the writing. R: The paper has been proofed by a native English speaker.  The inconsistencies have been corrected.

Reviewer 2 Report

The examined developmental disabilities are different among them, so it is expected that the parents level of stress could be different related to different disabilities. Could  this side of the problem be depeer investigated?

This study is focused on a multicultural muslim country like Iran: it could be interesting for western readers know more about the organization of health care for families with disabled children in this country. Could be some notes inserted?

Author Response

1. The examined developmental disabilities are different among them, so it is expected that the parents level of stress could be different related to different disabilities. Could  this side of the problem be depeer investigated?

R: This study does indeed provide an initial investigation of this relationship and we have highlighted further research that could be undertaken to explore the relationship more deeply. 

2. This study is focused on a multicultural muslim country like Iran: it could be interesting for western readers know more about the organization of health care for families with disabled children in this country. Could be some notes inserted?

R: A section has been added that outlines services the families received in Iran –see section 1.2. 

Reviewer 3 Report

see attached 

Author Response

1. The article “Parental satisfaction with caregiving across the life 3 span to their children with developmental disabilities: A cross-sectional study in Iran” is of great interest. The research is very stimulating; it contains new scientific knowledge and provides comprehensive information for further development of this productive line of research.

R: Thank you for this affirmation.

2. However, I have a major concern that may preclude the document for publication. The objectives part of the introduction seems a bit weak for me. Authors should clearly expose in a scientific way what are the specific objectives and hypothesis of their study based in previous literature.

R: We have rewritten the research questions and noted that they derive from the literature review presented in the introduction and that they address the gaps identified in the review.

3. Author should check for grammatical errors in word and sentence through the whole text. For instance: “The Cronbach alphas” is incorrect and should be revised.

R: The paper has been proofed by a native English speaker.  Cronbach’s alpha is used.

4. With regards to the participants’ section, did authors check for relevant variables like IQ or socioeconomic level. It seems reasonable that those parents with a higher socioeconomical status and more economical possibilities approach disability in a different way. If authors did not check for this, it should be noted in the limitations section.

R: Information on IQ was not available to us.  Homeownership was used as a proxy for socio-economic status of families. 

5. Also, authors should clarify if the participants were just one (mother or father) of the parents or if they recruited both father and mother of the same child. This seem an aspect that could affect the results.

R: We have clarified that only one parent per household took part in the study.

6. Besides this general comment, I consider that the research is stimulating and contribute valuable information about parents’ perception on a relevant matter as it is developmental disabilities of their children.

R: Thanks again for your kind words. 

Round 2

Reviewer 1 Report

The author has carefully addressed the reviewer's comments, but there are still some minor problems:

Page 3, line 8: The Cronbach’s alpha for these subscales were 0.883. It should be was 0.883.

Table 3: The correlation matrix measures the correlation between 5 measures, thus it should be 5 by 5. Why the authors only present 4 by 4 table?

Table 4: Row 5. Need to adjust the space between 'Stress – Child interaction'

Author Response

Thank you for the comments and suggestions, the following changes have been done: 

Page 3, line 8: The Cronbach’s alpha for these subscales were 0.883. It should be was 0.883. The sentence now reads. "The Cronbach's alphas ... were"

Table 3: The correlation matrix measures the correlation between 5 measures, thus it should be 5 by 5. Why the authors only present 4 by 4 table?   
As is standard practice, we omitted the correlations between the same variables which are 1.00 in all cases.

Table 4: Row 5. Need to adjust the space between 'Stress – Child interaction' 
The spaces on the table have been adjusted.